# Multi-Mention Learning for Reading Comprehension with Neural Cascades

**Swabha Swayamdipta**[*]
Language Technologies Institute
Carnegie Mellon University
Pittsburgh, PA 15213, USA
swabha@cs.cmu.edu

**Ankur P. Parikh & Tom Kwiatkowski**
Google Research
New York, NY 10011, USA
{aparikh,tomkwiat}@google.com

## Abstract

Reading comprehension is a challenging task, especially when executed across longer or across multiple evidence documents, where the answer is likely to re-occur. Existing neural architectures typically do not scale to the entire evidence, and hence, resort to selecting a single passage in the document (either via truncation or other means), and carefully searching for the answer within that passage. However, in some cases, this strategy can be suboptimal, since by focusing on a specific passage, it becomes difficult to leverage multiple mentions of the same answer throughout the document. In this work, we take a different approach by constructing lightweight models that are combined in a cascade to find the answer. Each submodel consists only of feed-forward networks equipped with an attention mechanism, making it trivially parallelizable. We show that our approach can scale to approximately an order of magnitude larger evidence documents and can aggregate information at the representation level from multiple mentions of each answer candidate across the document. Empirically, our approach achieves state-of-the-art performance on both the Wikipedia and web domains of the TriviaQA dataset, outperforming more complex, recurrent architectures.

## 1 Introduction

Reading comprehension, the task of answering questions based on a set of one more documents, is a key challenge in natural language understanding. While data-driven approaches for the task date back to Hirschman et al. (1999), much of the recent progress can be attributed to new large-scale datasets such as the CNN/Daily Mail Corpus (Hermann et al., 2015), the Children's Book Test Corpus (Hill et al., 2015) and the Stanford Question Answering Dataset (SQuAD) (Rajpurkar et al., 2016). These datasets have driven a large body of neural approaches (Wang & Jiang, 2016; Lee et al., 2016; Seo et al., 2016; Xiong et al., 2016, *inter alia*) that build complex deep models typically driven by long short-term memory networks (Hochreiter & Schmidhuber, 1997). These models have given impressive results on SQuAD where the document consists of a single paragraph and the correct answer span is typically only present once. However, they are computationally intensive and cannot scale to large evidence texts. Such is the case in the recently released TriviaQA dataset (Joshi et al., 2017), which provides as evidence, *entire* webpages or Wikipedia articles, for answering independently collected trivia-style questions.

So far, progress on the TriviaQA dataset has leveraged existing approaches on the SQuAD dataset by truncating documents and focusing on the first 800 words (Joshi et al., 2017; Pan et al., 2017). This has the obvious limitation that the truncated document may not contain the evidence required to answer the question[1]. Furthermore, in TriviaQA there is often useful evidence spread *throughout* the supporting documents. This cannot be harnessed by approaches such as Choi et al. (2017) that greedily search for the best 1-2 sentences in a document. For example, in Fig.1 the answer does not appear in the first 800 words. The first occurrence of the answer string is not sufficient to answer the question. The passage starting at token 4089 does contain all of the information required to infer

---

[*]Work done during internship at Google NY.
[1]Even though the answer string itself might occur in the truncated document.

the answer, but this inference requires us to resolve the two complex co-referential phrases in 'In the summer of that year they got married in a church'. Access to other mentions of Krasner and Pollock and the year 1945 is important to answer this question.

| Question | : | Which US artist married Lee Krasner in 1945? |
|---|---|---|
| **Answer** | : | Jackson Pollock ; Pollock ; Pollock, Jackson |
| **Document** | : | Wikipedia entry for *Lee Krasner* (Excerpts shown) |

| Start | Passage |
|---|---|
| 952 | She lost interest in their usage of hard-edge geometric style after her relationship with **Pollock** began. |
| 3364 | **Pollock**'s influence. |
| 3366 | Although many people believe that Krasner stopped working in the 1940s in order to nurture **Pollock**'s home life and career, she never stopped creating art. |
| 4084 | Relationship with Jackson **Pollock** |
| 4089 | Lee Krasner and Jackson **Pollock** established a relationship with one another in 1942 after they both exhibited at the McMillen Gallery. Krasner was intrigued by his work and the fact she did not know who he was since she knew many abstract painters in New York. She went to his apartment to meet him. *By 1945, they moved to The Springs on the outskirts of East Hampton. In the summer of that year, they got married in a church with two witnesses present.* |
| 4560 | While she married **Pollock** in a church, Krasner continued to identify herself as Jewish but decided to not practice the religion. |

Figure 1: Example from TriviaQA in which multiple mentions contain information that is useful in inferring the answer. Only the italicized phrase completely answers the question (Krasner could have married multiple times) but contains complex coreference that is beyond the scope of current natural language processing. The last phrase is more easy to interpret but it misses the clue provided by the year 1945.

In this paper we present a novel cascaded approach to extractive question answering (§3) that can accumulate evidence from an order of magnitude more text than previous approaches, and which achieves state-of-the-art performance on all tasks and metrics in the TriviaQA evaluation. The model is split into three levels that consist of feed-forward networks applied to an embedding of the input. The first level submodels use simple bag-of-embeddings representations of the question, a candidate answer span in the document, and the words surrounding the span (the context). The second level submodel uses the representation built by the first level, along with an attention mechanism (Bahdanau et al., 2014) that aligns question words with words in the sentence that contains the candidate span. Finally, for answer candidates that are mentioned multiple times in the evidence document, the third level submodel aggregates the mention level representations from the second level to build a single answer representation. At inference time, predictions are made using the output of the third level classifier only. However, in training, as opposed to using a single loss, all the classifiers are trained using the multi-loss framework of Al-Rfou et al. (2016), with gradients flowing down from higher to lower submodels. This separation into submodels and the multi-loss objective prevents adaptation between features (Hinton et al., 2012). This is particularly important in our case where the higher level, more complex submodels could subsume the weaker, lower level models c.f. Al-Rfou et al. (2016).

To summarize, our novel contributions are

- a non-recurrent architecture enabling processing of longer evidence texts consisting of simple submodels
- the aggregation of evidence from multiple mentions of answer candidates at the representation level
- the use of a multi-loss objective.

Our experimental results (§4) show that all the above are essential in helping our model achieve state-of-the-art performance. Since we use only feed-forward networks along with fixed length window representations of the question, answer candidate, and answer context, the vast majority of computation required by our model is trivially parallelizable, and is about $45\times$ faster in comparison to recurrent models.

## 2 RELATED APPROACHES

Most existing approaches to reading comprehension (Wang & Jiang, 2016; Lee et al., 2016; Seo et al., 2016; Xiong et al., 2016; Wang et al., 2017; Kadlec et al., 2016, *inter alia*) involve using recurrent neural nets (LSTMs (Hochreiter & Schmidhuber, 1997) or memory nets (Weston et al., 2014)) along with various flavors of the attention mechanism (Bahdanau et al., 2014) to align the question with the passage. In preliminary experiments in the original TriviaQA paper, Joshi et al. (2017) explored one such approach, the BiDAF architecture (Seo et al., 2016), for their dataset. However, BiDAF is designed for SQuAD, where the evidence passage is much shorter (122 tokens on an average), and hence does not scale to the entire document in TriviaQA (2895 tokens on an average); to work around this, the document is truncated to the first 800 tokens.

Pointer networks with multi-hop reasoning, and syntactic and NER features, have been used recently in three architectures — Smarnet (Chen et al., 2017b), Reinforced Mnemonic Reader (Hu et al., 2017) and MEMEN (Pan et al., 2017) for both SQuAD and TriviaQA. Most of the above also use document truncation .

Approaches such as Choi et al. (2017) first select the top (1-2) sentences using a very coarse model and then run a recurrent architecture on these sentences to find the correct span. Chen et al. (2017a) propose scoring spans in each paragraph with a recurrent network architecture separately and then take taking the span with the highest score.

Our approach is different from existing question-answering architectures in the following aspects. First, instead of using one monolithic architecture, we employ a cascade of simpler models that enables us to analyze larger parts of the document. Secondly, instead of recurrent neural networks, we use only feed-forward networks to improve scalability. Third, our approach aggregates information from different mentions of the same candidate answer at the representation level rather than the score level, as in other approaches (Kadlec et al., 2016; Joshi et al., 2017). Finally, our learning problem also leverages the presence of several correct answer spans in the document, instead of considering only the first mention of the correct answer candidate.

## 3 MODEL

For the reading comprehension task (§3.1), we propose a cascaded model architecture arranged in three different levels (§3.2). Submodels in the lower levels (§3.3) use simple features to score candidate answer spans. Higher level submodels select the best answer candidate using more expensive attention-based features (§3.4) and by aggregating information from different occurrences of each answer candidate in the document (§3.5). The submodels score all the potential answer span candidates in the evidence document[2], each represented using simple bags-of-embeddings. Each submodel is associated with its own objective and the final objective is given by a linear combination of all the objectives (§3.6). We conclude this section with a runtime analysis of the model (§3.7).

### 3.1 TASK

We take as input a sequence of question word embeddings $\mathbf{q} = \{\mathbf{q_1} \ldots \mathbf{q_m}\}$, and document word embeddings $\mathbf{d} = \{\mathbf{d_1} \ldots \mathbf{d_n}\}$, obtained from a dictionary of pre-trained word embeddings.

Each candidate answer span, $\mathbf{s} = \{\mathbf{d_{s_1}} \ldots \mathbf{d_{s_o}}\}$ is a collection of $o \leq l$ *consecutive* word embeddings from the document, where $l$ is the maximum length of a span. The set of all candidate answer spans is $\mathbf{S} := \{\mathbf{s_i}\}_{i=1}^{nl}$. Limiting spans to length $l$ minimally affects oracle accuracy (see Section §4) and allows the approach to scale linearly with document size.

---

[2]We truncate extremely long documents (§4), but with a truncation limit $10\times$ longer than prior work.

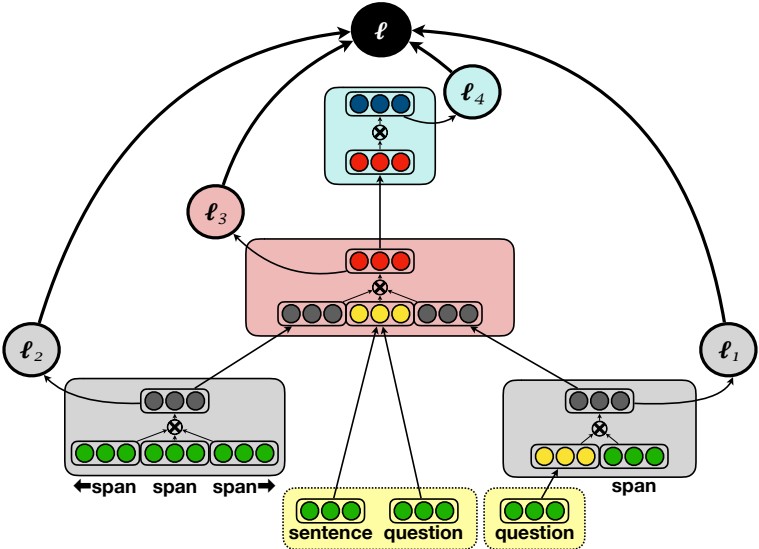

Figure 2: Cascaded model for reading comprehension. Input vectors are shown in green. Yellow rounded squares with dotted borders correspond to attention computation and attended vectors are shown in yellow. Submodels are shown in rounded squares with solid borders and the corresponding objectives are shown in color-coded circles. Level 1 submodels are in grey, level 2 in red and level 3 in blue. The **ffnn** operator is shown by the cross symbol within each submodel. The final objective, shown in the top black circle, is a linear interpolation of submodel objectives.

Since the same spans of text can occur multiple times in each document, we also maintain the set of unique answer candidate spans, $\mathbf{u} \in S_u$, and a mapping between each span and the unique answer candidate that it corresponds to, $\mathbf{s} \twoheadrightarrow \mathbf{u}$. In TriviaQA, each answer can have multiple alternative forms, as shown in Fig.1. The set of correct answer strings is $S^*$ and our task is to predict a single answer candidate $\hat{\mathbf{u}} \in S$.

## 3.2 OVERVIEW OF META-ARCHITECTURE

We first describe our meta-architecture, which is a collection of simple submodels $M_k(\cdot)$ organized in a cascade. The idea of modeling separate classifiers that use complementary features comes from Al-Rfou et al. (2016) who found this gave considerable gains over combining all the features into a single model for a conversational system. As shown in Figure 2, our architecture consists of two submodels $M_1$, $M_2$ at the first level, one submodel $M_3$ at the second, and one submodel $M_4$ at the third level. Each submodel $M_k$ returns a score, $\phi_s^{(k)}$ as well as a vector, $\mathbf{h}_s^{(k)}$ that is fed as input to the submodel at the next level.

$$\phi_s^{(1)}, \mathbf{h}_s^{(1)} := M_1(\mathbf{q}, \mathbf{d}, \mathbf{s}) \; \forall s \in S \qquad \phi_s^{(2)}, \mathbf{h}_s^{(2)} := M_2(\mathbf{q}, \mathbf{d}, \mathbf{s}) \; \forall \mathbf{s} \in S \qquad \textbf{LEVEL 1}$$

$$\phi_s^{(3)}, \mathbf{h}_s^{(3)} := M_3(\mathbf{q}, \mathbf{d}, \mathbf{s}, \mathbf{h}_s^{(1)}, \mathbf{h}_s^{(2)}) \; \forall s \in S \qquad \textbf{LEVEL 2}$$

$$\phi_u^{(4)}, \mathbf{h}_u^{(4)} := M_4(\mathbf{q}, \mathbf{d}, \mathbf{s}, \{\mathbf{h}_s^{(3)}\}_{\mathbf{s} \twoheadrightarrow \mathbf{u}}) \; \forall \mathbf{s} \in S_u \qquad \textbf{LEVEL 3}$$

Using their respective scores, $\phi_s^{(k)}$, the models $M_1...M_3$ define a distribution over all spans, while $M_4$ uses $\phi_{\mathbf{u}}^{(4)}$ to define a distribution over all unique candidates, as follows:

$$p^{(k)}(\mathbf{s}|\mathbf{q}, \mathbf{d}) = \frac{\exp \phi_{\mathbf{s}}^{(k)}}{\sum_{\mathbf{s}' \in S} \exp \phi_{\mathbf{s}'}^{(k)}} \quad k \in \{1, 2, 3\} \qquad p^{(4)}(\mathbf{u}|\mathbf{q}, \mathbf{d}) = \frac{\exp \phi_{\mathbf{u}}^{(4)}}{\sum_{\mathbf{u}' \in S_u} \exp \phi_{\mathbf{u}'}^{(4)}} \quad (1)$$

In training, our total loss is given by an interpolation of losses for each of $M_1, .., M_4$. However, during inference we make a single prediction, simply computed as $\hat{\mathbf{u}} = \arg\max_{\mathbf{u} \in S_{\mathbf{u}}} \phi_{\mathbf{u}}^{(4)}$.

### 3.3 LEVEL 1: AVERAGED BAGS OF EMBEDDINGS

The first level is the simplest, taking only bags of embeddings as input. This level contains two submodels, one that looks at the span and question together (§3.3.1), and another that looks at the span along with the local context in which the span appears (§3.3.2). We first describe the span representations used by both.

**Span Embeddings:**  We denote a span of length $o$ as a vector $\mathbf{s}$, containing

- averaged document token embeddings, and
- a binary feature $\gamma_{qs}$ indicating whether the spans contains question tokens

$$\tilde{\mathbf{s}} = [\frac{1}{o}\sum_{j=1}^{o} \mathbf{d}_{s_j}; \gamma_{qs}]$$

.

The binary feature $\gamma_{qs}$ is motivated by the question-in-span feature from Chen et al. (2017a), we use the question-in-span feature, motivated by the observation that questions rarely contain the answers that they are seeking and it helps the model avoid answers that are over-complete — containing information already known by the questioner.

#### 3.3.1 QUESTION + SPAN ($M_1$)

The question + span component of the level 1 submodel predicts the correct answer using a feed-forward neural network on top of fixed length question and span representations. The span representation is taken from above and we represent the question with a weighted average of the question embeddings.

**Question Embeddings:**  Motivated by Lee et al. (2016) we learn a weight $\delta_{q_i}$ for each of the words in the question using the parameterized function defined below. This allows the model to learn to focus on the important words in the question. $\delta_{q_i}$ is generated with a two-layer feed-forward net with rectified linear unit (ReLU) activations (Nair & Hinton, 2010; Glorot et al., 2011),

$$\mathbf{h}_{q_i} = \mathbf{ReLU}\{\mathbf{U}\{\mathbf{ReLU}\{\mathbf{Vq_i} + \mathbf{a}\}\} + \mathbf{b}\}$$
$$= \mathbf{ffnn}_q(\mathbf{q_i})$$
$$\delta_{q_i} = \mathbf{w}^T\mathbf{h}_{q_i} + \mathbf{z}$$
$$= \mathbf{linear}_q(\mathbf{h_{q_i}})$$

where $\mathbf{U}$, $\mathbf{V}$, $\mathbf{w}$, $\mathbf{z}$, $\mathbf{a}$ and $\mathbf{b}$ are parameters of the feed-forward network. Since all three submodels rely on identically structured feed-forward networks and linear prediction layers, from now on we will use the abbreviations **ffnn** and **linear** as shorthand for the functions defined above.

The scores, $\delta_{q_i}$ are normalized and used to generate an aggregated question vector $\tilde{\mathbf{q}}$ as follows.

$$\tilde{\mathbf{q}} = \sum_{i=1}^{m} \frac{\exp \delta_{q_i} \mathbf{q_i}}{\sum_{j=1}^{m} \exp \delta_{q_j}}$$

Now that we have representations of both the question and the span, the question + span model computes a hidden layer representation of the question-span pair as well as a scalar score for the span candidate as follows:

$$\mathbf{h}_s^{(1)} = \mathbf{ffnn}_{qs}([\tilde{\mathbf{s}}; \tilde{\mathbf{q}}; \gamma_{qs}]), \quad \phi_{s_i}^{(1)} = \mathbf{linear}_{qs}(\mathbf{h}_s^{(1)})$$

where $[\mathbf{x}; \mathbf{y}]$ represents the concatenation of vectors $\mathbf{x}$ and $\mathbf{y}$.

#### 3.3.2 SPAN + SHORT CONTEXT ($M_2$)

The span + short context component builds a representation of each span in its local linguistic context. We hypothesize that words in the left and right context of a candidate span are important for the modeling of the span's semantic category (e.g. person, place, date).

**Context Embeddings:** We represent the $K$-length left context $\mathbf{c_s^L}$, and right context $\mathbf{c_s^R}$ using averaged embeddings

$$\mathbf{c}_s^L = \frac{1}{K} \sum_{j=1}^{K} \mathbf{d}_{s_{1-j}}, \qquad \mathbf{c}_s^R = \frac{1}{K} \sum_{j=1}^{K} \mathbf{d}_{s_{o+j}}$$

and use these to generate a span-in-context representation $\mathbf{h}_s^{(2)}$ and answer score $\phi_s^{(2)}$

$$\mathbf{h}_s^{(2)} = \mathbf{ffnn}_c([\mathbf{s}; \mathbf{c}_s^L; \mathbf{c}_s^R; \gamma_{qs}]), \quad \phi_s^{(2)} = \mathbf{linear}_c(\mathbf{h}_s^{(2)}).$$

## 3.4 LEVEL 2: ATTENDING TO THE CONTEXT ($M_3$)

Unlike level 1 which builds a question representation independently of the document, the level 2 submodel considers the question in the context of each sentence in the document. This level aligns the question embeddings with the embeddings in each sentence using neural attention (Bahdanau et al., 2014; Lee et al., 2016; Xiong et al., 2016; Seo et al., 2016), specifically the attend and compare steps from the decomposable attention model of Parikh et al. (2016). The aligned representations are used to predict if the sentence could contain the answers. We apply this attention to sentences, not individual span contexts, to prevent our computational requirements from exploding with the number of spans. Subsequently, it is only because level 2 includes the level 1 representations $\mathbf{h}_s^{(1)}$ and $\mathbf{h}_s^{(2)}$ that it can assign different scores to different spans in the same sentence.

**Sentence Embeddings:** We define $\mathbf{g}_s = \{\mathbf{d}_{g_{s,1}} \dots \mathbf{d}_{g_{s,G}}\}$ to be the embeddings of the $G$ words of the sentence that contains $\mathbf{s}$. First, we measure the similarity between every pair of question and sentence embeddings by passing them through a feed-forward net, $\mathbf{ffnn}_{\text{att}_1}$ and using the resulting hidden layers to compute a similarity score, $\eta$. Taking into account this similarity, attended vectors $\bar{\mathbf{q}}_{\mathbf{i}}$ and $\bar{\mathbf{d}}_{g_{s,j}}$ are computed for each question and sentence token, respectively.

$$\eta_{ij} = \mathbf{ffnn}_{\text{att}_1}(\mathbf{q_i})^T \mathbf{ffnn}_{\text{att}_1}(\mathbf{d}_{g_{s,j}})$$

$$\bar{\mathbf{q}}_{\mathbf{i}} = \sum_{j=1}^{\ell} \frac{\exp \eta_{ij}}{\sum_{k=1}^{\ell} \exp \eta_{ik}} \mathbf{d}_{g_{s,j}} \qquad \qquad \bar{\mathbf{d}}_{g_{s,j}} = \sum_{i=1}^{m} \frac{\exp \eta_{ij}}{\sum_{k=1}^{m} \exp \eta_{kj}} \mathbf{q_i}$$

The original vector and its corresponding attended vector are concatenated and passed through another feed-forward net, $\mathbf{ffnn}_{\text{att}_2}$ the final layers from which are summed to obtain a question-aware sentence vector $\bar{\mathbf{g}}_s$, and a sentence context-aware question vector, $\bar{\mathbf{q}}$.

$$\bar{\mathbf{q}} = \sum_{i=1}^{m} \mathbf{ffnn}_{\text{att}_2}([\mathbf{q_i}; \bar{\mathbf{q}}_{\mathbf{i}}]) \qquad \qquad \bar{\mathbf{g}}_s = \sum_{j=1}^{\ell} \mathbf{ffnn}_{\text{att}_2}([\mathbf{d}_{g_{s,j}}; \bar{\mathbf{d}}_{g_{s,j}}])$$

Using these attended vectors as features, along with the hidden layers from level 1 and the question-span feature, new scores and hidden layers are computed for level 2:

$$\mathbf{h}_s^{(3)} = \mathbf{ffnn}_{L_2}([\mathbf{h}_s^{(1)}; \mathbf{h}_s^{(2)}; \bar{\mathbf{q}}; \bar{\mathbf{g}}_s; \gamma_{qs}]), \quad \phi_s^{(3)} = \mathbf{linear}_{L_2}(\mathbf{h}_s^{(3)})$$

## 3.5 LEVEL 3: AGGREGATING MULTIPLE MENTIONS ($M_4$)

In this level, we aggregate information from all the candidate answer spans which occur multiple times throughout the document. The hidden layers of every span from level 2 (along with the question-in-span feature) are passed through a feed-forward net, and then summed if they correspond to the same unique span, using the $s \twoheadrightarrow u$ map. The sum, $\mathbf{h}_u$ is then used to compute the score and the hidden layer[3] for each unique span, $u$ in the document.

$$\mathbf{h}_u^{(4)} = \mathbf{ffnn}_{L_3}\Big(\sum_{s \in u} \mathbf{ffnn}_{\text{agg}}([\mathbf{h}_s^{(3)}; \gamma_{qs}])\Big), \quad \phi_s^{(4)} = \mathbf{linear}_{L_3}(\mathbf{h}_u^{(4)})$$

---

[3]The hidden layer in level 3 is used only for computing the score $\phi_u^{(4)}$, mentioned here to preserve consistency of notation.

## 3.6 OVERALL OBJECTIVE AND LEARNING

To handle distant supervision, previous approaches use the first mention of the correct answer span (or any of its aliases) in the document as gold (Joshi et al., 2017). Instead, we leverage the existence of multiple correct answer occurrences by maximizing the probability of all such occurrences. Using Equation 1, the overall objective, $\ell(\mathbf{U}_*, \mathbf{V}_*, \mathbf{w}_*, \mathbf{z}_*, \mathbf{a}_*, \mathbf{b}_*)$ is given by the total negative log likelihood of the correct answer spans under all submodels:

$$-\sum_{k=1}^{3} \lambda_k \log \sum_{\hat{\mathbf{s}} \in \mathbf{S}^*} p^{(k)}(\hat{\mathbf{s}}|\mathbf{q}, \mathbf{d}) - \lambda_4 \log \sum_{\hat{\mathbf{u}} \in \mathbf{S}^*} p^{(4)}(\hat{\mathbf{u}}|\mathbf{q}, \mathbf{d})$$

where $\lambda_1, .., .\lambda_4$ are hyperparameters, such that $\sum_{i=1}^{4} \lambda_i = 1$, to weigh the contribution of each loss term.

## 3.7 COMPUTATIONAL COMPLEXITY

We briefly discuss the asymptotic complexity of our approach. For simplicity assume all hidden dimensions and the embedding dimension are $\rho$ and that the complexity of matrix$(\rho \times \rho)$-vector$(\rho \times 1)$ multiplication is $O(\rho^2)$. Thus, each application of a feed-forward network has $O(\rho^2)$ complexity. Recall that $m$ is the length of the question, $n$ is the length of the document, and $l$ is the maximum length of each span. We then have the following complexity for each submodel:

**Level 1 (Question + Span)** : Building the weighted representation of each question takes $O(m\rho^2)$ and running the feed forward net to score all the spans requires $O(nl\rho^2)$, for a total of $O(m\rho^2 + nl\rho^2)$.

**Level 1 (Span + Short Context)** : This requires $O(nl\rho^2)$.

**Level 2** : Computing the alignment between the question and each sentence in the document takes $O(n\rho^2 + m\rho^2 + nm\rho)$ and then scoring each span requires $O(nl\rho^2)$. This gives a total complexity of $O(nl\rho^2 + nm\rho)$, since we can reasonably assume that $m < n$.

**Level 3** : This requires $O(nl\rho^2)$.

Thus, the total complexity of our approach is $O(nl\rho^2 + mn\rho)$. While the $nl$ and $nm$ terms can seem expensive at first glance, a key advantage of our approach is that each sub-model is trivially parallelizable over the length of the document ($n$) and thus very amenable to GPUs. Moreover note that $l$ is set to 5 in our experiments since we are only concerned about short answers.

## 4 EXPERIMENTS AND RESULTS

### 4.1 TRIVIAQA DATASET

The TriviaQA dataset (Joshi et al., 2017) contains a collection of 95k trivia question-answer pairs from several online trivia sources. To provide evidence for answering these questions, documents are collected independently, from the web and from Wikipedia. Performance is reported independently in either domain. In addition to the answers from the trivia sources, aliases for the answers are collected from DBPedia; on an average, there are 16 such aliases per answer entity. Answers and their aliases can occur multiple times in the document; the average occurrence frequency is almost 15 times per document in either domain. The dataset also provides a subset on the development and test splits which only contain examples determined by annotators to have enough evidence in the document to support the answer. In contrast, in the full development and test split of the data, the answer *string* is guaranteed to occur, but not necessarily with the evidence needed to answer the question.

### 4.2 EXPERIMENTAL SETTINGS

**Data preprocessing:** All documents are tokenized using the NLTK[4] tokenizer. Each document is truncated to contain at most 6000 words and at most 1000 sentences (average the number of

---

[4]http://www.nltk.org

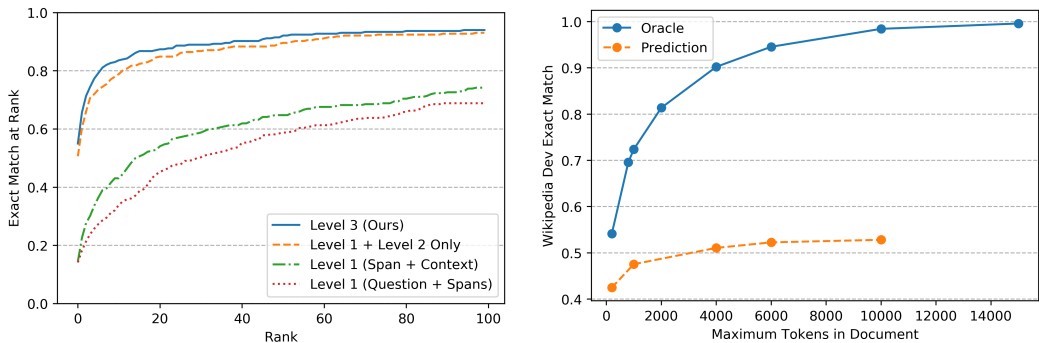

Figure 3: Analysis of model predictions. Left: Performance of the top $k$ predictions of different models on the human-validated Wikipedia development set. Right: Effect of truncation on performance. *Oracle* indicates the maximum possible performance for that truncation level.

sentences per document in Wikipedia is about 240). Sentences are truncated to a maximum length of 50 (avg sentence length in Wikipedia is about 22). Spans only up to length $l = 5$ are considered and cross-sentence spans discarded — this results in an oracle exact match accuracy of 95% on the Wikipedia development data. To be consistent with the evaluation setup of Joshi et al. (2017), for the Wikipedia domain we create a training instance for every question (with all its associated documents), while on the web domain we create a training instance for every question-document pair.

**Hyperparameters:** We use GloVe embeddings (Pennington et al., 2014) of dimension 300 (trained on a corpus of 840 billion words) that are not updated during training. Each embedding vector is normalized to have $\ell_2$ norm of 1. Out-of-vocabulary words are hashed to one of 1000 random embeddings, each initialized with a mean of 0 and a variance of 1. Dropout regularization (Srivastava et al., 2014) is applied to all ReLU layers (but not for the linear layers). We additionally tuned the following hyperparameters using grid search and indicate the optimal values in parantheses: network size (2-layers, each with 300 neurons), dropout ratio (0.1), learning rate (0.05), context size (1), and loss weights ($\lambda_1 = \lambda_2 = 0.35$, $\lambda_3 = 0.2$, $\lambda_4 = 0.1$). We use Adagrad (Duchi et al., 2011) for optimization (default initial accumulator value set to 0.1, batch size set to 1). Each hyperparameter setting took 2-3 days to train on a single NVIDIA P100 GPU. The model was implemented in Tensorflow (Abadi et al., 2016).

|  | Wikipedia | | | | Web | | | |
|  | Full | | Verified | | Full | | Verified | |
|  | EM | $F_1$ | EM | $F_1$ | EM | $F_1$ | EM | $F_1$ |
|---|---|---|---|---|---|---|---|---|
| BiDAF Seo et al. (2016); Joshi et al. (2017) | 40.33 | 45.91 | 44.17 | 50.52 | 40.73 | 47.05 | 48.63 | 55.07 |
| Smarnet Chen et al. (2017b) | 42.41 | 48.84 | 50.51 | 55.90 | 40.87 | 47.09 | 51.11 | 55.98 |
| Reinforced Mnemonic Reader Hu et al. (2017) | 46.90 | 52.90 | 54.50 | 59.50 | 46.70 | 52.90 | 57.00 | 61.50 |
| Leaderboard Best (10/24/2017) | 48.64 | 55.13 | 53.42 | 59.92 | 50.56 | 56.73 | **63.20** | **67.97** |
| Neural Cascades (Ours) | **51.59** | **55.95** | **58.90** | **62.53** | **53.75** | **58.57** | **63.20** | 66.88 |

Table 1: TriviaQA results on the test set. EM stands for Exact Match accuracy.

## 4.3 RESULTS

Table 1 presents our results on both the full test set as well as the verified subsets, using the exact match (EM) and $F_1$ metrics. Our approach achieves state-of-the-art performance on both the

Wikipedia and web domains outperforming considerably more complex models [5] . In the web domain, except for the verified $F_1$ scores, we see a similar trend. Surprisingly, we outperform approaches which use multi-layer recurrent pointer networks with specialized memories (Chen et al., 2017b; Hu et al., 2017)[6].

|  | Wikipedia Dev (EM) |
|---|---|
| 3-Level Cascade, Multi-loss (Ours) | **52.18** |
| 3-Level Cascade, Single Loss | 43.48 |
| 3-Level Cascade, Combined Level 1 ** | **52.25** |
| Level 1 + Level 2 Only | 46.52 |
| Level 1 (Span + Context) Only | 19.75 |
| Level 1 (Question + Span) Only | 15.75 |

Table 2: Model ablations on the full Wikipedia development set. For row labeled **, explanation provided in Section §4.3.

Table 2 shows some ablations that give more insight into the different contributions of our model components. Our final approach (3-Level Cascade, Multi-loss) achieves the best performance. Training with only a single loss in level 3 (3-Level Cascade, Single Loss) leads to a considerable decrease in performance, signifying the effect of using a multi-loss architecture. It is unclear if combining the two submodels in level 1 into a single feed-forward network that is a function of the question, span and short context (3-Level Cascade, Combined Level 1) is significantly better than our final model. Although we found it could obtain high results, it was less consistent across different runs and gave lower scores on average (49.30) compared to our approach averaged over 4 runs (51.03). Finally, the last three rows show the results of using only smaller components of our model instead of the entire architecture. In particular, our model without the aggregation submodel (Level 1 + Level 2 Only) performs considerably worse, showing the value of aggregating multiple mentions of the same span across the document. As expected, the level 1 only models are the weakest, showing that attending to the document is a powerful method for reading comprehension. Figure 3 (left) shows the behavior of the $k$-best predictions of these smaller components. While the difference between the level 1 models becomes more enhanced when considering the top-$k$ candidates, the difference between the model without the aggregation submodel (Level 1 + Level 2 Only) and our full model is no longer significant, indicating that the former might not be able to distinguish between the best answer candidates as well as the latter.

**Effect of Truncation**: The effect of truncation on Wikipedia in Figure 3 (right) indicates that more scalable approaches that can take advantage of longer documents have the potential to perform better on the TriviaQA task.

**Multiple Mentions**: TriviaQA answers and their aliases typically reoccur in the document (15 times per document on an average). To verify whether our model is able to predict answers which occur frequently in the document, we look at the frequencies of the predicted answer spans in Figure 4 (left). This distribution follows the distribution of the gold answers very closely, showing that our model learns the frequency of occurrence of answer spans in the document.

**Speed**: To demonstrate the computational advantages of our approach we implemented a simple 50-state bidirectional LSTM baseline (without any attention) that runs through the document and predicts the start/end positions separately. Figure 4 (right) shows the speedup ratio of our approach compared to this LSTM baseline as the length of the document is increased (both approaches use a P100 GPU). For a length of 200 tokens, our approach is about $8\times$ faster, but for a maximum length of 10,000 tokens our approach is about $45\times$ faster, showing that our method can more easily take advantage of GPU parallelization.

## 4.4 ANALYSIS

We observe the following phenomena in the results (see Table 3) which provide insight into the benefits of our architecture, and some directions of future work.

---

[5]TriviaQA allows private submissions, but we are only able to compare with results that are made public and/or published before our submission.

[6]We cannot compare to MEMEN (Pan et al., 2017) in this table since they provide only development set results; our Wikipedia dev EM is 52.18 compared to their 43.16.

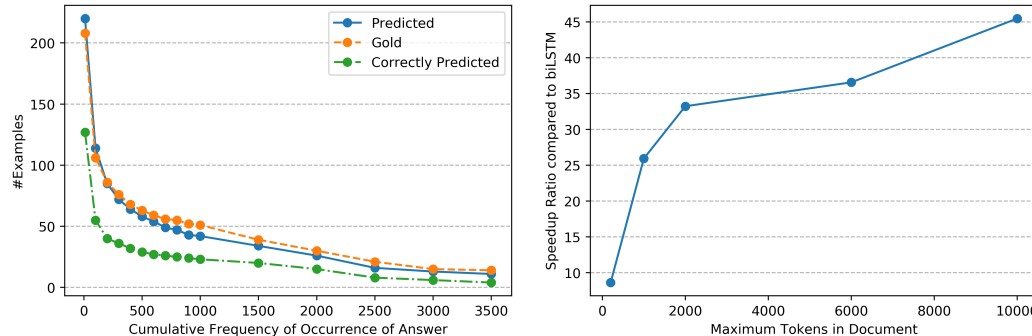

Figure 4: Left: Distribution of answer frequencies in the document, for model predictions (blue), correct model predictions (green) and true answers (orange). Right: Speedup ratio of our approach compared to a vanilla biLSTM that predicts start/end positions independently. As the document length increases, our approach can better take advantage of GPU parallelization.

| Question | Answer | Level 3 (ours) | Level 1 + 2 | Level 1 Span+Context | Question+Span | Phenomena |
|---|---|---|---|---|---|---|
| Which famous novelist also wrote under the pseudonym Richard Bachman ? | Stephen King | **Stephen King** | **Stephen King** | **Stephen King** | Lemony Snicket | Submodels which consider more context do better than the level 1 model which only considers the question and the span, out of context. |
| Terry Molloy , David Gooderson and Julian Bleach have all portrayed which villain in the UK television series Dr Who ? | Davros | **Davros** | Doctor Who | The Borgias | Dalek Caan | Davros is present a few times across the different evidence documents, which level 3 is able to aggregate and predict correctly. Doctor Who is the most frequently occurring entity across all evidence documents. Nevertheless, the aggregator (level 3) model refrains from selecting it. |
| Which villain , played by Richard Kiel , appeared in two James Bond movies , 'The Spy Who Loved Me ' and 'Moonraker ' ? | Jaws | **Jaws** | Lee Falk | Thunderball | Stavro Blofeld | Jaws occurs 45 times across all the evidence documents, whereas Thunderball occurs 25 times, Stavro Blofeld twice and Lee Falk once. |
| What name is given to a substance that accelerates a chemical reaction without itself being affected ? | Catalysts | 1.**Catalysts** 2.Inhibitors 3.Catalyst | 1.Inhibitors 2.**Catalysts** 3.**Catalysts** | 1.Joseph Proust 2.Oxidation 3.Double Arrow | 1.Molybd. Dioxide 2.Chlorine 3.Chlorine | Level 2 contains the correct answer multiple times further down its top predictions list (ranks shown), but is unable to combine these mentions. |
| High Willhays is the highest point of what National Park ? | Dartmoor | High Willhays | High Willhays | **Dartmoor** | Goblet of Fire | Lower levels get the prediction right, but not the upper levels. Model predicts entities from the question. |

Table 3: Example predictions from different levels of our model. Evidence context and aggregation are helpful for model performance. The model confuses between entities of the same type, particularly in the lower levels.

**Aggregation helps** As motivated in Fig 1, we observe that aggregating mention representations across the evidence text helps (row 3). Lower levels may contain, among the top candidates, multiple mentions of the correct answer (row 4). However, since they cannot aggregate these mentions, they tend to perform worse. Moreover, level 3 does not just select the most frequent candidate, it selects the correct one (row 2).

**Context helps** Models which take into account the context surrounding the span do better (rows 1-4) than the level 1 (question + span) submodel, which considers answer spans completely out of context.

**Entity-type confusion** Our model still struggles to resolve confusion between different entities of the same type (row 4). Context helps mitigate this confusion in many cases (rows 1-2). How-

ever, sometimes the lower levels get the answer right, while the upper levels do not (row 5) — this illustrates the value of using a multi-loss architecture with a combination of models.

Our model still struggles with deciphering the entities present in the question (row 5), despite the question-in-span feature.

## 5 CONCLUSION

We presented a 3-level cascaded model for TriviaQA reading comprehension. Our approach, through the use of feed-forward networks and bag-of-embeddings representations, can handle longer evidence documents and aggregated information from multiple occurrences of answer spans throughout the document. We achieved state-of-the-art performance on both Wikipedia and web domains, outperforming several complex recurrent architectures.

### ACKNOWLEDGMENTS

We thank Mandar Joshi for help with the TriviaQA dataset and the leaderboard. We also thank Minjoon Seo, Luheng He, Dipanjan Das, Michael Collins, Chris Clark and Luke Zettlemoyer for helpful discussions and feedback. Finally, we thank anonymous reviewers for their comments.

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
