# OpenReview forum: "Multi-Mention Learning for Reading Comprehension with Neural Cascades"
_ICLR.cc/2018/Conference — Accept (Poster)_

### Official Review · AnonReviewer2 · 2017-11-21
**Intuitive model for scaling question answering**

**Rating:** 7
**Confidence:** 4

**Review:**

The authors present a scalable model for questioning answering that is able to train on long documents. On the TriviaQA dataset, the proposed model achieves state of the art results on both domains (wikipedia and web). The formulation of the model is straight-forward, however I am skeptical about whether the results prove the premise of the paper (e.g. multi-mention reasoning is necessary). Furthermore, I am slightly unconvinced about the authors' claim of efficiency. Nevertheless, I think this work is important given its performance on the task.

1. Why is this model successful? Multi-mention reasoning or more document context?
I am not convinced of the necessity of multi-mention reasoning, which the authors use as motivation, as shown in the examples in the paper. For example, in Figure 1, the answer is solely obtained using the second last passage. The other mentions provide signal, but does not provide conclusive evidence. Perhaps I am mistaken, but it seems to me that the proposed model cannot seem to handle negation, can the authors confirm/deny this? I am also skeptical about the computation efficiency of a model that scores all spans in a document (which is O(N^2), where N is the document length). Can you show some analysis of your model results that confirm/deny this hypothesis?

2. Why is the computational complexity not a function of the number of spans?
It seems like the derivations presents several equations that score a given span. Perhaps I am mistaken, but there seems to be n^2 spans in the document that one has to score. Shouldn't the computational complexity then be at least O(n^2), which makes it actually much slower than, say, SQuAD models that do greedy decoding O(2n + nm)?

Some minor notes
- 3.3.1 seems like an attention computation in which the attention context over the question and span is computed using the question. Explicitly mentioning this may help the reading grasp the formulation.
- Same for 3.4, which seems like the biattention (Seo 2017) or coattention (Xiong 2017) from previous squad work.
- The sentence "We define ... to be the embeddings of the l words of the sentence that contains s." is not very clear. Do you mean that the sentence contains l words? It could be interpreted that the span has l words.
- There is a typo in your 3.7 "level 1 complexity": there is an extra O inside the big O notation.

---

> ### Author Response · Authors · 2017-12-08
> **Addressing AnonReviewer2 Concerns**
>
> Thank you for your comments! We will revise the paper based on your feedback but we would like to clarify some aspects beforehand:
>
> Success of the model:
> Our model benefits from both multi-mention reasoning and more document context. The ablation in Table 2 shows that without the Level 3 multi-mention aggregation, model performance drops from 52.18% to 46.52%. The only purpose of the level 3 model is to do aggregation across multiple mentions, and therefore this shows that multi-mention reasoning helps our model significantly. More document context allows the upper bound of the dev EM under our approach to be 92%, compared to the 83% in the baseline method from Joshi et. al. (2017). We can make this more clear in our revision.
>
> Computational efficiency:
> We only allow spans up to a length l (where l=5). Therefore our computational complexity is O(nl) and not O(n^2). Moreover, our method trivially parallelizes across the length of the document, unlike recurrent network based approaches.
>
> Negation:
> Our model does not handle negation specifically but negation is not typically a key aspect of existing reading comprehension tasks (as it is in sentiment analysis for instance).
>
> Attention:
> We will clarify the attention section and cite (Seo et al. 2017, Xiong et al. 2017). Unlike their approach, we use the word embeddings as input to the attention, not the LSTM states as they do.

---

### Official Review · AnonReviewer3 · 2017-11-28

**Rating:** 5
**Confidence:** 4

**Review:**

This paper proposes a method that scales reading comprehension QA to large quantities of text with much less document truncation than competing approaches. The model also does not consider the first mention of the answer span as gold, instead formulating its loss function to incorporate multiple mentions of the answer within the evidence. The reported results were state-of-the-art(*) on the TriviaQA dataset at the time of the submission deadline. It's interesting that such a simple model, relying mainly on (weighted) word embedding averages, can outperform more complex architectures; however, these improvements are likely due to decreased truncation as opposed to bag-of-words architectures being superior to RNNs.

Overall, I found the paper interesting to read, and scaling QA up to larger documents is definitely an important research direction. On the other hand, I'm not quite convinced by its experimental results (more below) and the paper is lacking an analysis of what the different sub-models are learning. As such, I am borderline on its acceptance.

* The TriviaQA leaderboard shows a submission from 9/24/17 (by "chrisc") that has significantly higher EM/F1 scores than the proposed model. Why is this result not compared to in Table 1?

Detailed comments:
- Did you consider pruning spans as in the end-to-end coreference paper of Lee et al., EMNLP 2017? This may allow you to avoid truncation altogether. Perhaps this pruning could occur at level 1, making subsequent levels would be much more efficient.
- How long do you estimate training would take if instead of bag-of-words, level 1 used a biLSTM encoder for spans / questions?
- What is the average number of sentences per document? It's hard to get an idea of how reasonable the chosen truncation thresholds are without this.
- In Figure 3, it looks like the exact match score is still increasing as the maximum tokens in document is increased. Did the authors try truncating after more words (e.g., 10k)?
- I would have liked to see some examples of questions that are answered correctly by level 3 but not by level 2 or 1, for example, to give some intuition as to how each level works.
- "Krasner" misspelled multiple times as "Kramer"

---

> ### Author Response · Authors · 2017-12-08
> **Addressing AnonReviewer3 Concerns**
>
> Thank you for your comments! We will revise our paper based on your feedback, particularly discussing more the contribution of each submodel and addressing your detailed comments. However, we would like to clarify some main points  below:
>
> The “chrisc” leaderboard submission:
> TriviaQA allows someone to submit privately and then make their result public later. Therefore while the web result for the chrisc model might have been submitted earlier, it was not publicly visible before the ICLR Oct 27 deadline and therefore was not included in our table. None of the other ICLR submissions we are aware of report this result either e.g. (https://openreview.net/forum?id=rJl3yM-Ab, https://openreview.net/forum?id=HJRV1ZZAW, https://openreview.net/pdf?id=B1twdMCab ) The paper itself was posted  on arXiv on 29th Oct (https://arxiv.org/abs/1710.10723) and only contained Web (and not Wikipedia) results. We would also like to point out that the chrisc model involves a two-stage pipeline, many layers of recurrent neural nets and is procedurally more involved than ours.
>
> Pruning spans:
> We did try pruning spans, based on the levels, i.e. level 1 considered all spans in the document (up to truncation) and level 2 considered the top K spans from level 1, and so on. However, we found this decreased the accuracy by ~4-5 points. This could be attributed to the lower levels pruning away good candidates, because they did not have access to more information, such as sentence context, and attention with the question. We will revise the paper to include these results.
>
> Using biLSTMs:
> Running a biLSTM over the entire document of length n to obtain span representations is not parallelizable over the document length and therefore would be much slower (unlike our approach which trivially parallelizes the attention computation over the O(nl) spans).
>
> Truncation stats:
> Truncating documents to contain at most 6000 tokens gives us an upper bound of 92% on dev EM in the Wikipedia domain (avg number of sentences being 141). 87% of documents in the Wikipedia dataset are fully covered under this truncation limit.
>
> We do not make any claims about the expressive power of BOW models vs RNNs. Our model performance can be attributed to the scalability of BOW architectures which can take advantage of longer documents, which RNN architectures are not well suited for. Furthermore, our other contributions (multi-loss + multi-mention learning) significantly boost the performance of these simple architectures as explored in the ablations in Table 2.

---

> ### Public Comment · ~Christopher_Clark1 · 2017-12-10
> **TriviaQA leaderboard**
>
> I am the author of the "chrisc" submission on the TriviaQA Leaderboard.
>
> I just wanted to comment and confirm that our result on the leaderboard was not made public until after the ICLR deadline. The date listed on the leader board reflects the time we uploaded our test results, but we did not make that result public until after we had finished writing the paper and completed the rest of our evaluations, which occurred shortly after the ICLR submission deadline.

---

> > ### Public Comment · ~Mandar_Joshi1 · 2017-12-11
> > **TriviaQA leaderboard**
> >
> > I'm the administrator for the TriviaQA leaderboard on Codalab. I second Chris' comment. The leaderboard allows private submissions. In addition, the date field for each entry on the leaderboard refers to the date of submission (and not the date it was made public).

---

### Official Review · AnonReviewer1 · 2017-11-28

**Rating:** 6
**Confidence:** 4

**Review:**

This paper proposes a lightweight neural network architecture for reading comprehension, which 1) only consists of feed-forward nets; 2) aggregates information from different occurrences of candidate answers, and demonstrates good performance on TriviaQA (where documents are generally pretty long).

Overall, I think it is a nice demonstration that non-recurrent models can work so well, but I also don’t find the results strikingly surprising. It is also a bit hard to get the main takeaway messages. It seems that multi-loss is important (highlight that!), summing up multiple mentions of the same candidate answers seems to be important (This paper should be cited: Text Understanding with the Attention Sum Reader Network https://arxiv.org/abs/1603.01547). But all the other components seem to have been demonstrated previously in other papers.

An important feature of this model is it is easier to parallelize and speed up the training/testing processes. However, I don’t see any demonstration of this in the experiments section.

Also, I am a bit disappointed by how “cascades” are actually implemented. I was expecting some sophisticated ways of combining information in a cascaded way (finding the most relevant piece of information, and then based on what it is obtained so far trying to find the next piece of relevant information and so on). The proposed model just simply sums up all the occurrences of candidate answers throughout the full document. 3-layer cascade is really just more like stacking several layers where each layer captures information of different granularity.

I am wondering if the authors can also add results on other RC datasets (e.g., SQuAD) and see if the model can generalize or not.

---

> ### Author Response · Authors · 2017-12-08
> **Addressing AnonReviewer1 Concerns**
>
> Thank you very much for your comments, we will revise the paper over the next few weeks based on your feedback. Below, we make some clarifications.
>
> Primary contributions of our work:
> Our work presents a novel and more scalable approach to question answering that is considerably different than the existing literature that is dominated by monolithic LSTM-based architectures. The takeaway messages regarding our approach are:
>
> 1. Multi-loss formulation, which to our knowledge has not been used in question answering, before. Empirically, this factors to a 10pt difference in the dev EM, as demonstrated in Table 2.
> 2. Aggregating multiple mentions of candidates at the representation level. We found this strategy to allow us to obtain high accuracy with simpler models when the multiple-mention assumption holds. Empirically, removing the aggregation level drops accuracy by 5.5 points in dev EM as shown in Table 2.
> 3. Unlike existing approaches, our model is trivially parallelizable in that we can process all the O(nl) spans in the document in parallel, allowing it to handle larger documents.
>
> We believe that since our approach can scale to much longer document spans with only 1 GPU attests to its scalability. We also provide asymptotic analysis of our runtime complexity.
>
> Cascades:
> We would like to point out in contrast to previous approaches that aggregate answers (e.g. Attention Sum Reader Network, which we will add a reference for), our method aggregates positions at the *representation* level (by adding vector representations of mentions), not at the score level. While we agree that there could be more complex ways of realizing cascades, we chose the simplest approach that would show the efficacy of such an idea. More sophisticated ways to combine information may not be compatible with the trivial parallelizability in our model.
>
> SQuAD dataset:
> While it could run on SQuAD, our model was specifically designed for a case that is different from SQuAD. In SQuAD, the evidence only consists of a paragraph (avg length 122 tokens, TriviaQA evidence is more than 20X longer) so scalability is not a concern. Furthermore, answers to SQuAD questions are almost always unique spans in the passage, hence many of our intuitions of multi-mention learning might not be relevant for this task.

---

### Author Response · Authors · 2018-01-04
**Revision of the draft based on reviewer comments**

We thank for the reviewers for their valuable feedback and have made the following main improvements to the paper:

-Content:
1. Speed comparison (Figure 4 - right): We compare the speed of our approach to a vanilla bi-LSTM on a GPU. Because of our approach is trivially parallelizable, it gets relatively much faster compared to the LSTM as the document length increases (reaching ~45x speedup for a truncation limit of 10K tokens).
2. Oracle statistics for truncation limit (Figure 3 - right): To justify our choice of truncation limit, we plot the oracle accuracy for various truncation thresholds.
3. Analysis of each submodel. We provide:
    a. Figure 3 - left: A quantitative analysis showing the performance of the top K results of each submodel
    b. Table 3: A table of examples showing predictions of each submodel, and cases where our aggregation model (level 3) is able to do more than other submodels.

-Writing:
1. Introduction: We have clarified the novelty of our approach:
    a. Multi-loss formulation, which to our knowledge has not been used in question answering, before. Empirically, this factors to a 10pt difference in the dev EM, as demonstrated in Table 2.
    b. Aggregating multiple mentions of candidates at the representation level. We found this strategy to allow us to obtain high accuracy with simpler models when the multiple-mention assumption holds. Empirically, removing the aggregation level drops accuracy by 5.5 points in dev EM as shown in Table 2.
    c. Unlike existing approaches, our model is trivially parallelizable in that we can process all the O(nl) spans in the document in parallel, allowing it to handle larger documents.

-Complexity: We have clarified that O(nl) is similar to O(n) since l = maximum span length and is restricted to 5, and is therefore not quadratic.

-Additional citations: Kadlec et al. 2016 (attention sum reader network), Seo et al. 2017, Xiong et al. 2017

-Analysis: Sample of predictions have been provided and analysed in Section 4.4.

-Experimental Results (Leaderboard): We believe we have addressed AnonReviewer3’s concerns (see the responses below), and also added a clarifying footnote to the paper.

---

### Decision · Program_Chairs · 2018-01-29
**ICLR 2018 Conference Acceptance Decision**

**Decision:**

Accept (Poster)

**Comment:**

The authors did a good job addressing reviewer concerns and analyzing and  testing their model on interesting datasets with convincing results.